



# The Landscape Fire Scars Database: mapping historical burned area and fire severity in Chile

Alejandro Miranda[1,2], Rayén Mentler[1], Ítalo Moletto-Lobos[3], Gabriela Alfaro[4], Leonardo Aliaga[1], Dana Balbontín[1],
Maximiliano Barraza[1], Susanne Baumbach[1], Patricio Calderón[1], Fernando Cárdenas[1], Iván Castillo[1], Gonzalo Contreras[1],
Felipe de la Barra[4], Mauricio Galleguillos[1,5], Mauro E. González[1,6,7], Carlos Hormazábal[1], Antonio Lara[1,6,8], Ian Mancilla[1],
Francisca Muñoz[1], Cristian Oyarce[1], Francisca Pantoja[1], Rocío Ramírez[1] and Vicente Urrutia[1]

[1]Center for Climate and Resilience Research (CR)2, Santiago, Chile.

[2]Laboratorio de Ecología del Paisaje y Conservación, Departamento de Ciencias Forestales, Universidad de La Frontera, Temuco, Chile.

[3]Image Processing Laboratory, Global Change Unit, University of Valencia, Valencia, Spain.

[4]Industrial Engineering Department, University of Chile, Santiago, Chile.

[5] Department of Environmental Sciences and Natural Resources, University of Chile, Santiago, Chile.

[6]Instituto de Conservación, Biodiversidad y Territorio, Facultad de Ciencias Forestales y Recursos Naturales, Universidad Austral de Chile, Valdivia, Chile.

[7]Center for Fire and Socioecosystem Resilience (FireSES), Universidad Austral de Chile, Valdivia, Chile

[8]Fundación Centro de los Bosques Nativos FORECOS, Valdivia, Chile

*Correspondence to*: Alejandro Miranda (alejandro.miranda@ufrontera.cl)

**Abstract.** Achieving a local understanding of fire regimes requires high resolution, systematic and dynamic databases. High-quality information can help to transform the evidence into decision-making in the context of rapidly changing landscapes, particularly considering that geographical and temporal patterns of fire regimes and their trends vary locally over time. Global fire scar products at low spatial resolutions are available, but high-resolution wildfire data, especially for developing
countries, is still lacking. Taking advantage of the Google Earth Engine (GEE) big-data analysis platform, we developed a flexible workflow to reconstruct individual burned areas and derive fire severity estimates for all reported fires. We tested our approach for historical wildfires in Chile. The result is the Landscape Fire Scars Database, a detailed and dynamic database that reconstructs 8,153 fires scars representing 66.6% of the country's officially recorded fires between 1985 and 2018. For each fire event the database contains the following information: (i) Landsat mosaic of pre- and post-fire images;
(ii) the fire scar in binary format; (iii) the remotely sensed estimated fire indexes (NBR, RdNBR), plus two vector files indicating (iv) the fire scar perimeter and (v) the fire scar severity reclassification. The Landscape Fire Scars Database for



Chile and GEE script (JavaScript) are publicly available. The framework developed for the database can be applied anywhere in the world, the only requirement being its adaptation to local factors such as data availability, fire regimes, land cover or land cover dynamics, vegetation recovery, and cloud cover.


## 1 Introduction

Wildfires as a natural phenomenon have been a key component of the terrestrial system for millions of years, shaping biome structure and composition, and influencing the Earth's system cycles. Human activity has dramatically modified natural

wildfire regimes and is now the main driver of their spatial and temporal patterns (Balch et al., 2017; Bowman et al., 2011). The changing fire regime has become an increasing threat to biodiversity (Kelly et al., 2020), agricultural and timber production (Stougiannidou et al., 2020; de la Barrera et al., 2018) and rural/peri-urban communities (Radeloff et al., 2018) as well as a major contributor to greenhouse gas emissions (Giglio et al., 2013). Recent estimates point to a global mean burned area of 337 to 423 Mha every year (Giglio et al., 2013, 2018). However, the geographical and temporal patterns of fire

regimes and their trends over time vary locally depending on the source of ignition (Ganteaume and Syphard, 2018), climate characteristics and their changes (Jolly et al., 2015; Duane et al., 2021), predominant land use and land cover (Butsic et al., 2015), railroad density (Amato et al., 2018) as well as firefighting and fire suppression and prevention capacity (Bowman et al., 2011; Moritz et al., 2014). Additionally, each natural or anthropogenic forcing factor differs in its impact on fire regime attributes (e.g., ignition, severity, burned area, intensity) across multiple spatial and temporal scales worldwide (Ager et al.,

2014; Balch et al., 2017; Fusco et al., 2016). An understanding of fire regimes at a local level requires high resolution, systematic and dynamic databases in order to transform the evidence into decision-making in these rapidly changing landscapes (Bowman et al., 2020).

Remote sensing provides pre-, during, and post-fire biophysical information necessary for conducting fire-risk assessment,

fire detection and monitoring, assessment of fire impacts, and follow-up of changes in land cover trends after fire occurrence (Szpakowski and Jensen, 2019). Recent public datasets and products have enabled a better understanding of global and regional wildfire patterns (Giglio et al., 2016, 2018; Schroeder et al., 2014). Although the principal active fire and burned area products contain information going back to the year 2000 (e.g., MODIS) with a spatial resolution in the best cases of more than 250 m (Chuvieco et al., 2018), there is still a lack of high-resolution wildfire data, especially for developing

countries (Chuvieco et al., 2019). Andela et al. (2019) created a global dataset for the period 2003 to 2016 that estimates the size, duration, and propagation rate of individual wildfires with a spatial resolution of 500 m using MODIS products. Likewise, Artés et al., (2019), also using MODIS products, developed a global dataset to analyze fire regimes and fire behavior based on ignition dates and daily burned areas for individual wildfires. The large discrepancies between local and global estimates of burned area occur mostly in the case of fires of less than 100 ha due to detection difficulties when using

coarse-resolution products (Roteta et al., 2019). This constitutes a significant barrier to the proper understanding of local





wildfire regimes, and highlights the need for a high-resolution wildfire database (Chuvieco et al., 2019). Recent efforts using Landsat images have led to the identification of annual burn probabilities per pixel from which a database with a 30 m spatial resolution has been constructed that reaches back to the 1980s, but this has been done only for developed countries such as the USA and Australia (Goodwin and Collett, 2014; Hawbaker et al., 2017). However, recent computational

advances and the free availability of satellite imagery catalogs provide a promising framework for mapping annual burned areas worldwide at a spatial resolution of 30 m, which would be a major step forward in high-resolution wildfire database generation (Long et al., 2019).

In the case of Chile, the fire regime has been described mainly on the basis of the public wildfire database maintained by the

Chilean Forest Service (CONAF), and with MODIS monthly burned area data used only in the most recent studies (de la Barrera et al., 2018; McWethy et al., 2018). Evidence regarding burned areas and fire frequency is derived from data with spatial resolutions between 500 m and 5 km (Gómez-González et al., 2019; González et al., 2018). From these large-scale datasets it has been determined that fire frequency is closely related to human footprint zones such as cities or other densely human-populated areas (Gómez-González et al., 2019; McWethy et al., 2018), roads (Miranda et al., 2020) and agricultural

or industrial forest plantation activities (Gómez-González et al., 2019; McWethy et al., 2018). However, burned area also strongly interacts with climatic conditions favorable to the spread of fires, especially warmer and dryer years associated with El Niño-Southern Oscillation, wet winters the year previous (Holz et al., 2017; Urrutia-Jalabert et al., 2018) and severe drought (González et al., 2018). Such conditions have been more prevalent/frequent in recent years, with increasing temperatures and a general reduction in precipitation reported for the area since 1980 and a prolonged megadrought since

2010 (Boisier et al., 2016; Garreaud et al., 2019). Fire ignition near human communities, favorable climatic conditions and a lack of landscape or fuel management lead to increased wildfire occurrence (Úbeda and Sarricolea, 2016). However, this large-scale understanding may still be insufficient, especially for local applications such as fire spread modeling, fire severity estimation, landscape planning and design, ecological impacts and ecosystem resilience, or national greenhouse gas emission estimation.


An excellent opportunity for developing countries to generate their own local and historical high-resolution databases of wildfire scars is provided by Google Earth Engine (GEE) (Long et al., 2019). GEE is an open cloud-computing platform for geospatial analysis that contains a public catalog of satellite images, topography, land covers and other environmental datasets (Gorelick et al., 2017). Taken advantage of this big-data analysis platform, we generate a detailed database of fire

scars in Chile through the development of a flexible workflow, enabling us to reconstruct individual burned areas and fire severity information for all reported historical fires. The result is our Landscape Fire Scars Database for Chile, which along with the GEE script (JavaScript) used to generate it are publicly available at https://www.pangaea.de/tok/6dcc6e08241c5076ef6bff47bbe73014308d4881 and https://code.earthengine.google.com/554027d16823525d890ab2f6c45167d9 respectively. This framework could be



implementable for any geographical area globally, requiring only that it be adapted to local conditions regarding seed data availability, fire regimes, land cover or land cover dynamics, vegetation recovery and cloud cover.

## 2 Data and methods

### 2.1 Study site

The approach we developed was applied to central and south-central Chile (29°S-43°5'S), a long stretch of territory encompassing ten of the country's administrative regions (Figure 1). Fire activity in Chile is concentrated in this area, where considerable changes in land use and land cover have been observed in recent decades (Miranda et al., 2017), associated with increased fire activity (González et al., 2018).

### 2.1 Data seeding


To construct our historical database of fire scars, we used a subset of the public wildfire database provided by CONAF (www.conaf.cl/conaf/seccion-stadisticas-historicas.html). For each recorded wildfire, CONAF indicates the geographic coordinates of the ignition points and the fire start and control dates and the burned area estimation (in ha). Given the image availability, quality and spatial resolution of the Landsat programs, we extracted data only for fires with a burned area of

more than 10 ha between 1985 and 2018 (N: 13,603). This original CONAF point dataset is included in our database's.

### 2.2 Fire scar generation

Our database was generated using JavaScript programming, GEE native language. The detailed workflow of the script

developed to create individual fire scars is shown in Figure 2. It consists of the following consecutive steps: (i) input data selection and identification; (ii) pre- and post-fire image elaboration; (iii) index, mask calculation, and vectorization; (iv) spatial and spectral filtering; and (v) output data generation and exporting. As noted earlier, we have made the GEE script available to all users as a tool that can be adapted to local conditions and used for permanent database updating. The code is available at https://code.earthengine.google.com/554027d16823525d890ab2f6c45167d9.


The input data in Step (i) must be in the form of point data with geographic coordinates representing the ignition point or a point within the burned area. The points must indicate the fire start date, the fire control date (fire spread ending date) and the estimated burned area. In the absence of the last two, we used the fire start date and a fixed burned area of 100 ha as seed values. The inputted seed data are converted into a list to processed and extracted individual fire scars. Around each input

point, a circular buffer area is created as a function of the estimated burned area, the precise dimensions given by $Buffer radius = log(burned area) * 2000$.

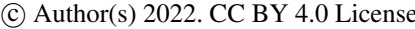

In Step (ii), two image collections (sets of images) are prepared for each wildfire, depending on the fire start date. We use the atmospherically corrected surface reflectance and orthorectified images from Landsat 5 (1984-2013), 7 (1999-) and 8

(2013-), with one image collection for a pre-fire condition and another for a post-fire condition, all of which are available in GEE. To avoid conflicts in mathematical operations for pre- and post-image collection generation, the date in day/month/year format is converted to Unix time format representing the number of milliseconds that have elapsed since January 1970. Based on the fire start and control dates, the respective image searches for both pre-fire post-fire events are each conducted for a period of 100 days. If this proves insufficient to get at least one image, the period can be extended up to

two years for a pre-fire event and six months for a post-fire event. Pixels of snow, clouds, and cloud shadows are excluded from each image on the basis of the pixel quality band provided by Landsat. For each image collection, we applied either the *mosaic* or the *median* reducer function to get a unique image of the landscape conditions at moments as close as possible before and after a fire event.

With the final pre- and post-mosaic images thus obtained, in Step (iii) we calculated all of the spectral indices (Table 1) used to identify the burned and unburned areas. The most widely used burned area index is the Normalized Burned Ratio (NBR) and its multitemporal form, the Delta Normalized Burn Ratio (dNBR) (Lentile et al., 2006; Fassnacht et al., 2021). These indexes reduce detection errors caused by shadows, water bodies, agricultural or tree harvesting, flooding and snowmelt (Chuvieco et al., 2019; Long et al., 2019). Other burned-area indexes have been proposed and a combination of them may

give the best results, but to discriminate between burned and unburned areas we opted for the Relative Delta Normalized Burn Ratio (RdNBR). This index has shown better results in Mediterranean areas (Miller and Thode, 2007).








Table 1: Description of spectral indexes and formulas used in the workflow.

| Index | Abbreviation | Formula | Usage | Reference |
|---|---|---|---|---|
| Normalized Difference Vegetation Index | NDVI | $\dfrac{\rho NIR - \rho RED}{\rho NIR + \rho RED}$ | Detects pre- and post-fire vegetation cover | (Rouse et al, 1974) |
| Normalized Burned Ratio | NBR | $\dfrac{\rho NIR - \rho SWIR2}{\rho NIR + \rho SWIR2}$ | Detects burned areas | (Key and Benson, 1999) |
| Delta Normalized Burn Ratio | Dnbr | $PreFireNBR - PostFireNBR$ | Detects changes in NIR and SWIR bands to identify burned area and fire severity | (Key and Benson, 1999) |
| Relative Delta Normalized Burn Ratio | RdNBR | $\dfrac{PreFireNBR - PostFireNBR}{\sqrt{\phantom{x}}}$ | Normalization of changes by pre-fire vegetation condition | (Miller and Thode, 2007) |

Step (iv) involved the selection of the RdNBR index value for each wildfire that best captures the burned area based on
visual interpretation. The raster mask of the burned area was converted to a vector format for spatial and spectral filtering
(Figure 1). By vectorizing the initially identified burned patches, spatial and spectral information could be added to each one
so that burned and unburned patches could be better distinguished using new criteria, thus diminishing commission errors.
This information included the mean NDVI both before and after the fire event, the Near Infrared (NIR) minimum value after
the event, and each patch's calculated area. We also calculated the NDVI in order to estimate several vegetation parameters
based on the red and infrared spectral bands (Table 1). The NDVI can be used to represent both the current state of, and
changes over time in, the composition, structure, and phenology of vegetation, as well as plant health and even burned
vegetation. (Helman, 2018; Pettorelli et al., 2005). Spatial filtering begins by defining an initial search distance to the
ignition point. The biggest patch within that distance is then identified and a new distance from this patch is defined. Only
the patches within this latter distance are considered. In this stage, polygons or patches that may cause commission errors are
eliminated from the areas counted as burned in the preliminary mask. They may include (a) water bodies with a pre-fire
mean NDVI of less than 0.1; (b) polygons or patches for which the pre-fire mean NDVI is less than 0.1 and therefore did not
contain vegetation, and other filtering criteria similar to those proposed by Long et al., (2019). Each polygon or patch
satisfies the filter criteria and has a minimum area of 0.3 ha is retained. The filter values can be changed to suit local
conditions.


Finally, in Step (v), once the fire scar is delimited, the event's severity is calculated from the RdNBR in a continuous raster
format and categorized based on the ranges proposed by Miller and Thode (2007). Our database also makes available the



pre- and post-fire NBR index for each image. Each fire scar and its severity are exported in vector and raster format, together with the multispectral corrected Landsat images of pre- and post-fire events and the RdNBR index. The vector data contains

information about the fire record, the calculated area and the spectral responses used for filtering. The output name of each vector and raster file is OBJECT (FireScar, Severity, ImgPre, ImgPost and RdNBR) +_ISO-REGION_ID +_u-THRESHOLD RdNBR VALUE +_START DATE, where ISO-REGION is the name of the administrative region based on the ISO 3166-2:CL norm, ID is the identification number of the evaluated fire, THRESHOLD VALUE is the numerical value of the RdNBR index used to separate burned and unburned areas. Finally, START DATE is the date used to find the

first image previous to the fire, which in most cases will be the same as the fire start date in the day/month/year format (e.g., FireScar_CL-RM_ID1920451_u330_19990215). A detailed description of each variable and its format is included as supplementary material in the database metadata.

**2.3 Fire scar evaluation**


We compared our fire scars with those generated by CONAF for the 2015-2016, 2016-2017 and 2017-2018 fire seasons. and published in Brull (2018). The author elaborated a manual digitalization of the fire scar perimeters using secondary information such as pre- and post-fire Landsat satellite images, dNBR index, Visible Infrared Imaging Radiometer Suite (VIIRS) active fire data, and Sentinel 2 images for high-resolution interpretation. The fire perimeters were defined as the

outer limit between the burned and unburned area in the landscape, but the unburned areas inside this perimeter were not discounted in the final fire scars. The author generated 194 fire scars, of which 78 coincided with those we reconstructed and were thus the ones used for making comparisons. The mean area of the 78 fire scars was 1,180 ha (min: 200, max: 12,250). In order to avoid confusion between fire events, the evaluation carried out for individual fires located at least 300 m from any other scar dating to the same season. The evaluation itself was based on the index proposed by Singh et al. (2015) that

compares two georeferenced polygons using the Closeness Index (D) as formulated in Eq. (1) below:

$$D(i,j) = \sqrt{\left(OverSegementation(i,j)\right)^2 + \left(UnderSegementation(i,j)\right)^2}$$

(1)

where i = reference polygon, j = segment polygon, OverSegmentation (i, j) = $1 - \frac{A_{(i,j)}}{A_{reference}(i)}$, UnderSegmentation (i, j) = $1 - \frac{A_{(i,j)}}{A_{segment}(j)}$, Aintersect (i, j) = common area between segment polygon j and corresponding reference polygon i, Areference (i) = Area of reference polygon i, and Asegment (j) = Area of segment polygon j

In order to normalize the values of D, we use the modification form $D_{norm} = 1 - \left(\frac{D}{\sqrt{(2)}}\right)$,



where $D_{norm}$ is the normalization of D values between 0 (no matching polygons) and 1 (perfectly matching polygons).

**2.4 Database quality control**

Even though the data generation process is done with standard and stable GEE scripts, the project's enormous scope could
lead to involuntary discordances in resulting files. A thorough revision was performed over approximately 140,000 files,
taking into account three major areas: (i) file and layer naming, file readability and type and amount of files per fire scar; (ii)
geographic locations and burned area related revision; and (iii) dates and season related revision. The approach was to define
several tests regarding relations between the content and attributes of the files in each area, that the whole dataset should
comply. The revision scripts were written in Python in the Google Collab environment, having direct access to the Google
Drive files generated by the GEE process. The tests were written for our resulting database but are generic in most terms and
assumptions and are available at https://github.com/cr2uchile/Quality_Control_FireScarCL. Some of these tests led to human
revision of the fires, either regenerating them or removing them from the firescar database, and other tests led to automated
fixes, like name change or attribute column and content changes in the vector files. The resulting database of 8153 fire scar
complies with the following statements:


- All fires have a unique identifier and 17 related files: Two satellite composite raster tif images that cover a domain larger
than the identified fire scar, that merge pre and a post images ( ImgPreF ..tif, ImgPosF ..tif), three raster tif images with the
shape of the fire scar that contains: zeroes where there is no fire scar identified and ones where there is (FireScar ..tif); zeroes
where there is no fire scar identified and severity index values (from 1 to 3) to identify the severity where there is a fire scar
(Severity ..tif) and the RdNBR value (float numbers) for the points where there is a fire scar (RdNBR ..tif). Finally two
vector Shapefile images that contain six files each (.shx, .shp, .dbf, .cpg, .fix, .prj) where one is the vectorized representation
of (FireScar ..shp) and the other is the vectorized representation of (Severity ..shp) with polygon and attributes information.
- For each set of resulting fire scar files, the ISO-REGION_ID corresponds to the region assigned by original CONAF point
dataset, and the START_DATE corresponds to the ignition point assigned by CONAF. This was preserved to better identify
the resulting fire scars with the seed database.
- All raster tif image files have the same area type and coordinate system. All pre and post-fire tif images have eight readable
bands.
- For each fire: the pre and a post-fire tif images have the same width and height dimensions and the exact geographic extent.
Also, their domain contains the firescar's ignition point and the resulting raster fire scar tif images (FireScar ..tif, Severity
..tif, RdNBR ..tif). The FireScar and Severity vector shapefiles files have consistent values in their attribute tables, and the
amount of polygons of the Severity vector image is equal or more than the amount of polygons of the FireScar vector image.
The dates in the attributes tables have format YYYY-MM-DD and the texts have UTF-8 encoding. The original fire names
with accented vowels and ñ, were replaced by the non-accented vowels and n, respectively.



## 3. Results

Using the data for all 12,250 fires recorded by CONAF between 1985 and 2018 with a burned area greater than 10 ha, we were able to reconstruct 8,153 fire scars, 66.56% of the total registered fires (Table 2, Figure 1). Suitable images were found for 35% of recorded fires for the period 1985-1994, 63% for 1995-2004, 82% for 2005-2014 and 93% for 2015-2018. The increasing trend evident in these percentages reflects how image availability has grown over time. Smaller numbers of suitable images were found for the country's southern regions (Los Ríos and Los Lagos), the wettest and coldest included in our study, where cloud cover is continuous for much of the year (Table 2).

Table 2: Regional and temporal distribution of fires and reconstructed fire scars. The administrative regions are those included in the study. The number of fires column indicates the total recorded by CONAF for which burned area was over 0.01 ha, R is the number of reconstructed fire scars contained in our database, and UR is the number of fire scars in the database that could not be reconstructed due to the unavailability of satellite images.

| Administrative region | Number of fires | Number of fires >10ha | Reconstructed fires scars > 10ha (%) | Total fire scars 1985-2018 | | 1985-1994 | | 1995-2004 | | 2005-2014 | | 2015-2018 | |
|---|---|---|---|---|---|---|---|---|---|---|---|---|---|
| | | | | Yes | No | Yes | No | Yes | No | Yes | No | Yes | No |
| Coquimbo | 1863 | 238 | 60.92 | 145 | 93 | 27 | 51 | 40 | 24 | 38 | 17 | 40 | 1 |
| Valparaíso | 31857 | 1784 | 80.38 | 1434 | 350 | 400 | 160 | 352 | 40 | 425 | 140 | 257 | 10 |
| Metropolitana | 15337 | 1109 | 85.75 | 951 | 158 | 208 | 79 | 252 | 42 | 261 | 33 | 230 | 4 |
| Ohiggins | 8249 | 1221 | 85.09 | 1039 | 182 | 240 | 93 | 251 | 56 | 365 | 26 | 183 | 7 |
| Maule | 14475 | 1419 | 65.89 | 935 | 484 | 103 | 290 | 199 | 118 | 393 | 52 | 240 | 24 |
| Ñuble and Bío-Bío | 77704 | 3248 | 58.07 | 1886 | 1362 | 124 | 775 | 473 | 375 | 712 | 171 | 577 | 41 |
| Araucanía | 31306 | 2369 | 57.41 | 1360 | 1009 | 30 | 458 | 346 | 356 | 424 | 131 | 560 | 64 |
| Los Ríos | 3680 | 339 | 35.99 | 122 | 217 | 8 | 154 | 41 | 48 | 33 | 10 | 40 | 5 |
| Los Lagos | 8416 | 523 | 53.73 | 281 | 242 | 5 | 100 | 53 | 111 | 143 | 27 | 80 | 4 |
| **Total** | **192,887** | **12,250** | **66.6** | **8,153** | **4,097** | **1,145** | **2,160** | **2,007** | **1,170** | **2,794** | **607** | **2,207** | **160** |

The total number of fires >0.01 ha exhibits a positive linear relationship with the total number of fires > 10 ha also recorded by CONAF between 1985 and 2018 ($R^2 = 0.86$). The number of recorded fires >10 ha and the number of reconstructed fire scars per region exhibits the same positive linear relationship ($R^2 = 0.92$), indicating that the distribution of the reconstructed data is regionally representative (Table 2, Figure 2). However, the pattern of relationships between recorded fires and reconstructed fire scars for the different regions varies from period to period. For 1985-1994 the relationship was weak ($R^2 = 0.1$) but had strengthened by 1995-2004 ($R^2 = 0.91$), and again for 2005-2014 ($R^2 = 0.93$) and 2015-2018 ($R^2 = 0.93$). The definitive version of our database is ordered by region and fire season to facilitate exploration and analysis, revealing, for



example, the high levels of fire activity areas near the coastal cities of Valparaíso and Concepción over the various decades (Figure 3).

For each of the 8,153 reconstructed fire scars, our database contains the following: (i) a Landsat mosaic of pre- and post-fire event images (.tif) with eight spectral bands: blue, green, red, NIR, SWIR1 and SWIR2, NDVI and NBR index (Figure 4); (ii) the raster of the fire scar in binary format (.tif), where 1 is the burned area and 0 the unburned area (Figure 4); and (iii) the RdNBR index, both in continuous values (.tif) and categorized by severity classification level, where 0 is unchanged, 1 is low severity, 2 is medium severity and 3 is high severity (Miller and Thode, 2007) (Figure 4). In addition, there are two

vector files (.shp) containing (iv) the fire scar perimeter and (v) the fire scar severity classification (Miller and Thode, 2007). Layers of information are assigned to each individual burned patch indicating its size, detected fire start and control dates, and spectral data. NBR bands are available for each image to enable reassessment of the fire scar and its severity. A detailed description of each variable and its format may be found in the database metadata.

## 3.1 Fire scar evaluation

We evaluated the fire scars reconstructed using our approach by contrasting them with the 78 scars derived from the official CONAF data that were suitable for making comparisons. A perfect match could not, of course, be expected given the differences in the two methodologies. One particularly crucial difference is that CONAF's fire scar digitalization includes

within the fire perimeter for each fire event patches that in fact were not burned. These patches constituted anywhere from 13.5% to 18.2% of the areas indicated as burned, depending on the fire season (Brull, 2018). Also, CONAF's digitalization was complemented by the agency's own fieldwork, which improved the detection of low severity fires or surface fire under the canopy. Nevertheless, the global accuracy result is 0.79. Examples of the comparisons of our reconstructed fire scars with CONAF data reported by (Brull, 2018) are shown in Figure 4 in together with the respective $D_{norm}$ index for each case.

## 3.2 Limitations and other observations regarding the Landscape Fire Scars Database

1. Our fire scar dataset does not represent all of the fires recorded in the 1985-2018 period.
2. The reconstructed fire scars are mainly concentrated in the last 20 years of that period, which may be related to the improvement over time in image availability.

3. Remotely sensed fire severity estimates the change in spectral response in the burned area and must be carefully treated in the analysis of the fires' ecological impact. Low severity or surface fire may be underestimated.
4. Due to the 16-day interval between Landsat images, one fire scar reconstructed from them may represent more than one fire event in neighboring areas experiencing multiple fires over that interval, especially in the case of originally independent





fire events that may have merged. Some fire scars in the database may be duplicated if they merged with another fire due to
their proximity in space and time. We include a notification in the database where this could have happened.

5. Commission errors may occur due to other land cover changes such as tree plantation clearcutting or harvesting on crops.

6. In certain cases, the inclusion of additional available images of pre- and post-fire events may help to improve the fire
scars.

**4. Data availability**

The Landscape fire scars dataset for Chile can be downloaded from the PANGAEA repository at
https://www.pangaea.de/tok/6dcc6e08241c5076ef6bff47bbe73014308d4881 (Miranda et al., 2022).

**5. Discussion and conclusions**

The creation of our Landscape Fire Scars Database for Chile makes publicly available for the first time a high-resolution
burned area product for the country. The georeferenced database is a multi-institutional effort containing information on
more than 8,000 fires events of more than 10 ha between 1984 and 2018. It contains data on fire scar area, perimeter, and
severity, which is accessible to the general public for analyzing future changes, improvements and new evaluations.
Furthermore, the methodology for generating these data was implemented in GEE so that others may replicate our approach
or apply it to other countries or cases where no openly accessible datasets are available. Public institutions and researchers
can take advantage of this framework to generate long-term time series of fire scars for any years of interest or just for one
particularly significant wildfire. The international community can replicate this workflow using national fire occurrence data
with the minimum required information or with recently released data on ignition coordinates, date, and fire duration for
more than 13 million individual fires worldwide that occurred between 2003 and 2016 (Andela et al., 2019). As a high-
resolution fire scar database, it should be of much help in conducting accurate and systematic evaluations of underlying
wildfire forces, impacts and recoveries, and delineating populations and biodiversity, public policy and informed territorial
decision making and planning (Chuvieco et al., 2019; Long et al., 2019; Stenzel et al., 2019).

Creating this database based on information distributed over an extensive territory on a national scale using a single method
presented diverse challenges as regards (i) historical image availability, (ii) land cover and land cover change dynamics, and
(iii) temporal image resolution and image cloud cover. In what follows, each of these issues is discussed in turn.

(i) Historical image availability



GEE (https://earthengine.google.com) provides free online access to original and corrected Landsat program data and products. Users do not need to download the images, and the analysis and image modification is also online, powered by Google servers (Gorelick et al., 2017). Image availability in the Landsat program is rather uneven across countries, with those in the developing world generally less well represented in terms of historical records. Nevertheless, the continuity of the image time series improves noticeably as the time period in question approaches the present. In the case of Chile, this pattern of improvement is clearly evidenced in the fire scare generation success rates we obtained for time periods since the mid-eighties (1985-1994: 35%, 1995-2004: 63%, 2005-2014: 82%, 2015-2018: 93%). This tendency must be considered when determining the time periods for reconstructing a database for any specific region. For example, according to the Landsat Global Archive Consolidation updates (Wulder et al., 2016), availability and usable image quality are lower for southern hemisphere high latitude regions (Huang et al., 2010; Stillinger et al., 2019; Viale et al., 2019).

(ii) Land cover change dynamics

Almost 90% of wildfire ignitions and burned areas worldwide are human-caused (Ganteaume and Syphard, 2018). As a result, many of these fires impact the wildland-urban interface, urban and rural settlements and productive regions (e.g., agricultural lands, tree plantations). Zones with high rates of land-use or land-cover changes may present some difficulties in fire scar and severity mapping. Remotely sensed burn area indexes are based on the abrupt change in the pre-fire spectral band values following a fire event. For example, NBR uses the near-infrared (NIR) and short wave infrared (SWIR) bands as proxies of photosynthetic productivity and water content of vegetated areas (Lentile et al., 2006; van Wagtendonk et al., 2004). Both parameters are affected by fire, so the greater is the temporal difference in the index, the greater was the event's severity. However, the spectral response of those bands may also be influenced by other factors. Forestry activity, especially tree plantation clearcutting, deforestation or harvesting on agricultural land, as well as the drying of annual grassland in the summer season, dried meadows, and the cultural practice of burning agricultural wastes may all act to confuse the spectral response for a given landscape, assimilating them to wildfire (Ghermandi et al., 2019). Another local consideration is the recovery rate of the vegetation. For example, in the tropics recovery is faster than in temperate areas, which could affect mapping of burned areas or fire intensity estimation depending on how much time has passed between fire occurrence and acquisition of a good quality satellite image (Chuvieco et al., 2019). Local topography may also complicate the process of distinguishing burned areas in mountain zones due to the increased presence of shadows, fog, or melting snow in cold or high-elevation areas (Huang et al., 2010; Stillinger et al., 2019; Viale et al., 2019). Therefore, local experience in landscape dynamics and practice is crucial to ensuring the generation of accurate databases and may constitute a basis for adapting the most commonly used burned area indexes to local realities.

(iii) Temporal image resolution and image cloud cover



Landsat images are widely used to study land cover changes and trends thanks to their spatial and temporal resolution (Soulard et al., 2016). However, the 16-day interval between images could be a major limitation. In regions with high fire activity, this can make it more difficult to identify individual fire scars and differentiate them from those produced on other days at neighboring locations. This means that a single final fire scar may in fact have been created by multiple fire events
occurring over the 16 days that converged or totally fused. The problem could be mitigated by using Sentinel-2 images (also available in GEE) for the earliest fire events given that the Sentinel-2 program is available from middle 2015 with a temporal resolution of 5-6 days and a pixel size of 10 m, although increasing spatial resolution may raise another issue in that it could result in underestimation of the influence of dead vegetation shadows on the spectral index signals (Fassnacht et al., 2021). The high temporal resolution could also be helpful in zones with high cloud cover such as are found in tropical and high
latitude or mountain areas (King et al., 2013).  For example, we can observe that the Landsat archive in Africa could reduce its number of images with less than 40% of cloud cover in a mean of more than 25% with much fewer images in the tropical zone of the Congo Basin (Roy et al., 2010).

The RdNBR index is able to differentiate burned area over a diverse range of climate and geographic conditions. No evident
pattern associated with the latitudinal or vegetation-type change was observed in applying the threshold value to identify scars. Different burned-land covers may have variable RdNBR values, but this relationship does not figure among the objectives for the present database development. In general, RdNBR performs wells when compared with field plots of severity. It is little influenced by the type of forest and is determined mainly by the fraction of consumed canopy cover (Cardil et al., 2019; Soverel et al., 2010; Fassnacht et al., 2021), demonstrating the index's high versatility. Nevertheless, the
task of assessing the performance of the severity classification is left to users of the database, and will depend on the local land cover context and field validations for identification of the best index. Our database does provide the NBR band for the images to facilitate comparison and evaluation of the dNBR and RdNBR indices.

The importance of the proposed database also stems from its value as a source of input for methods based on artificial
intelligence (AI) aimed at automating the process of generating new fire scars. AI techniques such as machine learning (ML), deep learning (DL), and especially the convolutional neural network (CNN) are increasingly being used for classification or object segmentation problems (Alzubaidi et al., 2021). The integration of such methods with remote sensing data is enabling the development of burned area detection models that use human-delimited wildfire perimeters as their training data set. Promising results have been achieved using uni- or multi-temporal images and different types of remote
sensing data to address the many open challenges in wildfire mapping and monitoring (Hu et al., 2021; Knopp et al., 2020; Pinto et al., 2021).

In conclusion, this present study makes, we believe, a significant contribution to the development of high-resolution methods for mapping fire scars and their temporal and spatial patterns. Our hope is that it will serve as a first step in an ongoing effort

to build and maintain an extensive, consistent database on forest fires in Chile that will drive scientific research and improvements in landscape management. Further study is needed to broaden the current state of knowledge on local conditions through standardized field surveys.

## 6. Acknowledgements


A.M., AL, MG, MGC, FM thank ANID/FONDAP/15110009, A.M. thanks the Agencia Nacional de Investigación y Desarrollo de Chile (ANID) Postdoctoral Fondecyt project 3210101 and FONDEF-IDEA ID20I10137. We also thank the support from the Complex Engineering Systems Institute PIA/BASAL AFB180003. M.E.G. thanks ANID/Fondecyt N° 1201528 and the Center for Fire and Socioecosistem Resilience (FireSES). The authors thank the Corporación Nacional
Forestal (CONAF) for providing the seeding and digitalized fire scars data for database evaluation. Special acknowledgments to Jordi Brull and the hundreds of anonymous firefighters and professionals who have observed, surveyed, and developed the CONAF official fire records since 1984. We also thanks to the Chilean Institute for Disaster Resilience (ITREND) for their support.


**Author contributions.** A.M., R.M., I.M., M.E.G., A.L., I.C., M.G., designed the study, database and found acquisition,. A.M. and R.M. managed the project and wrote the original draft with contributions from all other authors. A.M, R.M., G.A., L.A, D.B., M.B., S.B., P.C., F.C., G.C., F. dlB., C.H, I.M., C.O., F.P., R.R. and V. U. made the image interpretation, data processing, development of data bases, and providing different input on the manuscript and the database. F.M. made the data
curation and database quality control.

Competing interests. The authors declare that they have no conflict of interest.







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




**Figures**



Figure 1: Detailed workflow for individual fire scar generation in © Google Earth Engine. See Table 1 for details on NBR and

RdNBR.



Earth System Science Data Discussions — Open Access




Figure 2: A. Geographic distribution of the fire scar database. B and C show examples of fire activity near the cities of Valparaíso and Concepción for different periods.

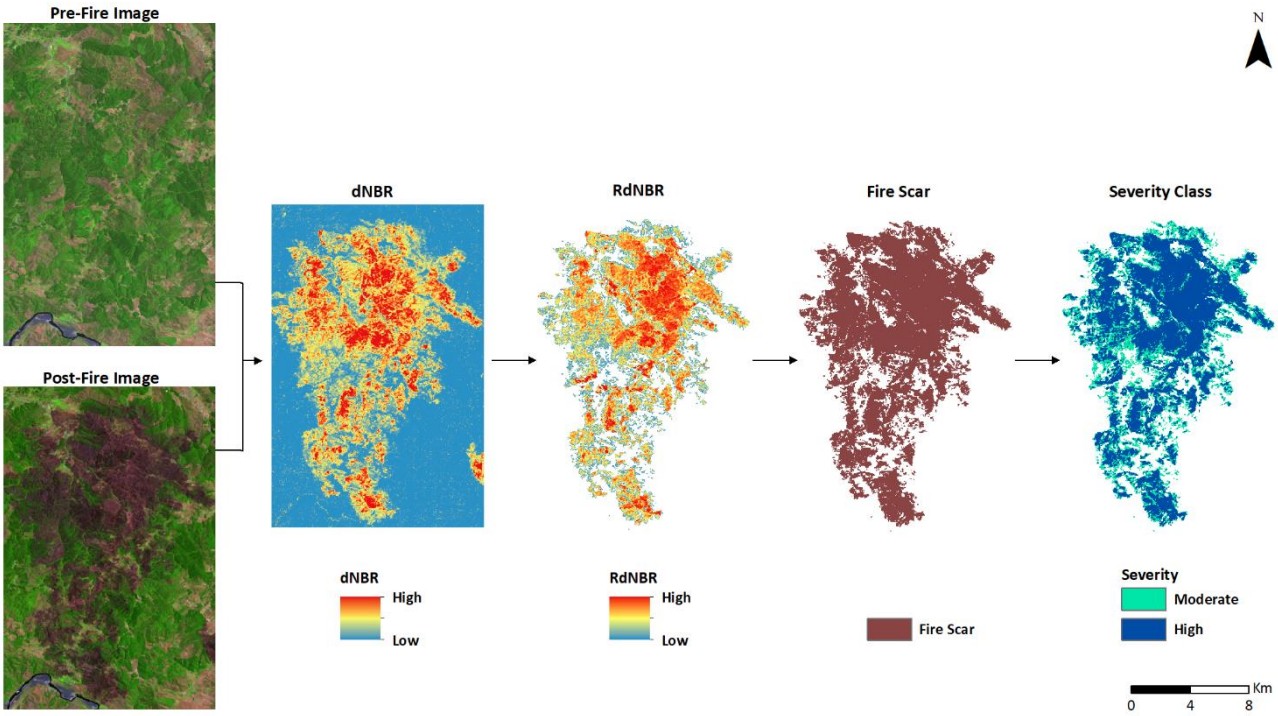

Figure 3: Database content for each reconstructed fire scar. See Table 1 for details on dNBR and RdNBR


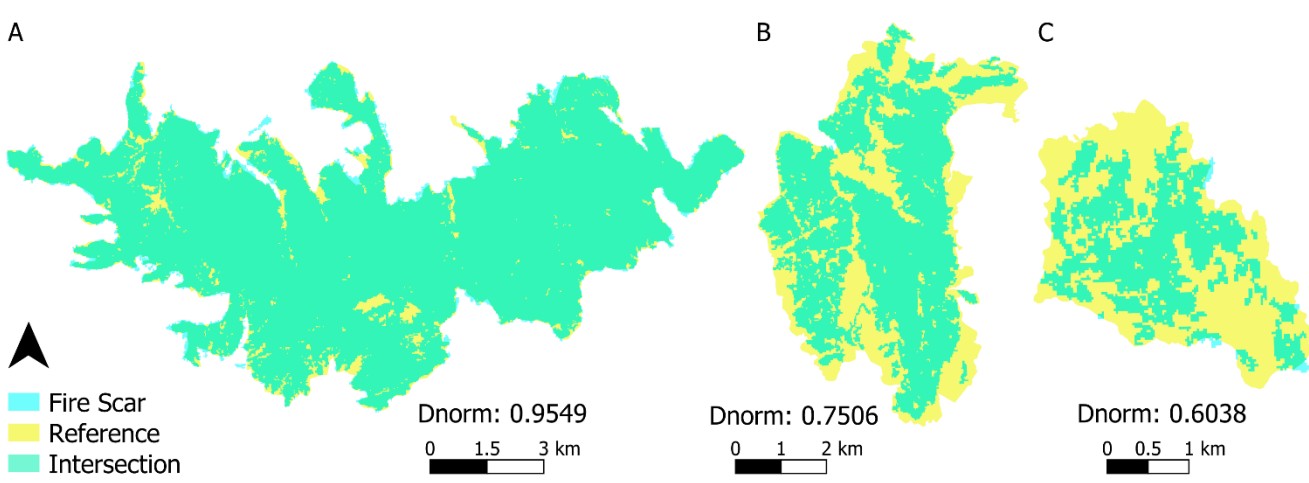

Figure 4: Evaluation of the fire scars. Shown are three examples comparing CONAF's fire scars with the images reconstructed using our Landscape Fire Scars Database methodology.