# Peer review of "The Landscape Fire Scars Database: mapping historical burned area and fire severity in Chile"

_Earth System Science Data, 2021_

## Author Response (AR1)

**RC1: Anonymous Referee #1**

The objective of the study is to reconstruct improved attributes of the historical fire database of Chile by leveraging Landsat imagery and Google Earth Engine (GEE) platform. The proposed workflow within GEE is well executed. The approach can semi-automatically generate different levels of fire products with homogeneous and unambiguous naming conventions. The introduction is well written. Accuracy assessment and validation of the generated products are also reasonable. Challenges of the approach and uncertainties of the results were acknowledged appropriately in the discussion. Figures and Tables are clearly presented. Availability of data sets and codes publicly is very nice.

The methodology and discussion sections are written well, however, it contains some ambiguities that could be clarified in the text.

1. The fire area cut-off threshold of 10 ha seems too high provided the 30 m resolution of Landsat imagery.

R: We agreed with the reviewer comment; this is why we add an explanation that justifies the cut-off threshold of 10 ha:

L116: "The 10 ha cut-off threshold was chosen since those fires represent more than 93% of the burned area according to the CONAF official information for 1985-2018 period. In addition, small fires are usually confounded with agricultural burning, a traditional cultural practice done by Chilean farmers"

2. The thresholds used for generating fire masks are not clear. For instance, it is not clear whether a global or local threshold was used. Provided the fire scar mask was generated individually, it is appropriate to use local threshold and conduct sensitivity test. The threshold can vary from one fire scar to another based on the condition at the time of image acquisition. Similarly, the sensitivity of thresholds used for spectral filtering could be elaborated a bit.

R: As the reviewer said, the threshold is defined by an interactive process. We include the sentence: L 173: "This is an interactive process (fire by fire) based on visual assessment of the best RdNBR value that delimitates each individual fire scar". The evaluation of threshold sensitivity is beyond the scope of the paper; however, the individual RdNBR is provided in the dataset for future assessment.

3. Chile seems to be maintaining an up-to-date historical fire point database (CONAF). A major setback for the workflow is likely transferability of the approach to the areas where there is a no or limited point database of historical fire. Alternatives or improvements that could be made to the workflow can be elaborated. For instance, an automation of the identification of thresholds (manually identifying threshold for each fire scar and severity can be challenging); could FIRMS database be useful at least starting 2000 for seeding the workflow?

R: Global datasets such as FIRMS take as the minimum size of pixel 375m per pixel (14.06 ha.), so it can be challenging to use this database for small fires. However, it can be used for medium or large fires. The following text as a new section has been added in the Discussion and conclusions section.

L426: "(iv) Seeding datasets

Studies in Chile have previously evaluated the performance of the global satellite burned area products showing good performance. An improvement of that's products can be done using as seed points the fires detected by global datasets in Chile or another area. Global datasets such as MODIS Collection 6 burned area data MCD64A1 (Giglio et al., 2009), FIRMS (Schroeder, Wilfrid and Giglio, Louis, 2018) and Global Fire Atlas (Andela et al., 2019; Giglio et al., 2018) can be used for evaluating the performance of our approach in medium and large fires and create new high-resolution datasets for mapping fire scars in different ecosystems and land covers."

4. How does the Landscape Fire Scars Database fit into other larger/global fire databases (e.g., Global Fire Atlas)? The data set could be relevant for studies at global scale in addition to its use cases like fire management in Chile.

R: Landscape Fire Scars Database is a data set for Chile but also a replicable workflow for creating high-resolution fire databases in other countries. In this context, our work can be used as an algorithm to improve the scale of fire scars of global data set or evaluate the accuracy assessment in hot-spot at MODIS or VIIRS at pixel scale. This answer is added in the Discussion and conclusions section proposed in Q3:

L426: "(iv) Seeding datasets

Studies in Chile have previously evaluated the performance of the global satellite burned area products showing good performance. An improvement of thats products can be done using as seed point the fires detected by global datasets in Chile or another area. Global datasets such as MODIS Collection 6 burned area data MCD64A1 (Giglio et al., 2009), FIRMS (Schroeder, Wilfrid and Giglio, Louis, 2018) and Global Fire Atlas (Andela et al., 2019; Giglio et al., 2018) can be used for evaluating the performance of our approach in medium and large fires and create new high-resolution datasets for mapping fire scars in different ecosystems and land covers."

**Specifics**

L125: "...must be in the form of point data". Is there any alternative when the fire database/record does not exist?

R: Yes, we include a new section in "5. Discussion and conclusions" addressing this point using global freely available datasets (L:426).

L125: "100 ha as seed value", explanation is needed for the choice.

R: This is only an initial value. It could be another initial value or then change it depending on the initial visual assessment of the fire. We clarify that this value is only for initial assessment in the text (L133).

L125: The reason for drawing a buffer around a point is not clear. Is it for extracting attributes for spectral filtering?

R: We explain in the text as follows:

L 135: Buffer area is explained because we may have only the fire's ignition point as a spatial reference. Therefore, it is necessary to explore near this point to find the limits of the fire scar. This could be an interactive process depending on the burned area or the shape of the fire scar.

L140: When was the 'mosaic' and 'reduce' function used? Does the approach decide which one to use based on some condition?

R: We applied the mosaic reducer for each image collection to get a unique image of the landscape conditions at moments as close as possible before and after a fire event.

We add the following text:

L151: When this reducer did not provide good quality pixels, we applied the median reducer instead. The median method for reducing image collections avoids extreme values by selecting the median value for each pixel.

L150: The suitability of the use of RdNBR in Chile needs to be justified if it was used for the first time.

R: The RdNBR was not used for the first time in Chile, the studies of Martini et al., (2020) and Fassnacht et al., (2021) used the index before. In any case we provide the RdNBR and the dNBR indexes in the dataset.

- Lorenzo Martini, Lorenzo Faes, Lorenzo Picco, Andrés Iroumé, Emanuele Lingua, Matteo Garbarino, Marco Cavalli. 2020. Assessing the effect of fire severity on sediment connectivity in central Chile. Science of The Total Environment https://doi.org/10.1016/j.scitotenv.2020.139006.
- Fabian Ewald Fassnacht, Ephraim Schmidt-Riese, Teja Kattenborn, Jaime Hernández. 2021. Explaining Sentinel 2 based dNBR and RdNBR variability with reference data from the bird's eye (UAS) perspective. International Journal of Applied Earth Observation and Geoinformation. https://doi.org/10.1016/j.jag.2020.102262.

Table 1: Fix the abbreviation "Dnbr". Value inside the square root is missing.

R: Fixed.

L170: Explicitly mention the "new criteria". It appears at L180, but it could be clearer.

R: We added new text to clarify the utility of this step in the workflow: L176,"The information added to the initial burned area patches could help to filter misclassified areas as burned areas, thus reducing commission errors"

L175: "initial search distance" what was the value? Is it the same for all fire scars?

R: We change the sentence to L180: "Spatial filtering begins by defining an initial search distance to the ignition point of 1 km, but it can be interactively defined or modified later"

L180: "0.3 ha is retained" is confusing. Is it a fire scar? How is it different from the 10 ha threshold?

R: We use all fires registered by CONAF with an estimated minimum fire area of 10 ha. However, one fire could be composed of more than one burned patch. We found that more than one burned patch can compose a unique fire event. To avoid misclassification in small area patches, we decide to filter them.

L210: "Closeness Index", does it considers the shape similarity between reference and mapped fire scar?

R: The Closeness index considers spatial matching between polygons but does not explicitly consider the shape similarity. However, we can presume that high values of the Closeness index also suggest high shape similarity.

**RC2: Anonymous Referee #2**

The paper contributes a new database for mapping historical burned areas and fire severity in Chile, called the Landscape Fire Scars Database. The manuscript itself is extremely thorough and comprehensive, and the dataset and codes are easily accessible for readers/users. There is a minor typo in line 229, Google Colab (not Collab). Overall, I found the engagement of the literature to be extensive and the motivation of the research to be well-justified. The conversation in the text about the benefits and limitations of the approach was written clearly and informatively. The figures were well made, and the GEE JavaScript codes ran without any issue. I was able to access the data without any issues as well. In general, I find this paper to be a wonderful contribution that will be particularly useful for other developing nations mapping similar landscape changes.

R: We appreciate the positive review.

**RC3: Anonymous Referee #3**

**[General comments]**

The objective of this article is to present and make available the "Landscape File Scars Database", a collection of historical fires in Chile built from officially reported fires and Landsat data. Availability of this data is an asset for researchers of various fields and to the fire-science community in particular. The latter will be glad to employ this source for the training or validation of BA regional or global products. The effort to create and make this database available it is a good example of a multi-institutional endeavour that countries developed to different degrees could focused on.

**[Specifical comments]**

Below I noted some points which could be addressed to improve the article (some of them are described more in detail later in the line-by-line comment section):

1)    It is recommended a deeper review of the CONAF existing fire/BA data be added: a section detailing the BA data CONAF products before the creation of this database, so the contribution of the new database is better understood. This would help in the comparison of the number of fires reconstructed (Table 2) that suggests there is a more accurate source available. Also, it would be beneficial to add detailed information about the source of the point database employed to reconstruct the fires.

R: We include the following paragraph in section 2.1 Data seeding to address the observations: L 109: "To construct our historical database of fire scars, we used a subset of the public wildfire database provided by Corporación Nacional Forestal (CONAF). This agency records and stores information on all fires (> 0.01 ha) regarding their location, date, causes, area affected by land use, date and time of first control and suppression of fire, among other variables. The georeferencing system used by CONAF until 2003 assigned each fire to the center of a 1x1 km alphanumeric grid, based on the subdivision of 1:50,000 scale maps of the Military Geographic Institute (IGM). After 2003, the location of each fire and estimation of their burned area began to be carried out with the help of a Global Positioning System (GPS)."

2)    The methodology employed to map burned areas is not totally clear; even though the code is available for GEE users, a nice feature that ensures reproducibility.  Firstly, it is not clear when 'mosaic' or 'median' reducers are employed to reduce imagecollections to images. If the 'median' were employed, this would reduce the noise but would soften the burned signal as well, an important feature for severity mapping. Secondly, the way an analyst define the threshold to map the burned area is not clearly defined - I guessed this is an interactive process based on visual assessment, however it is not clearly detailed. In addition, when describing the mapping process in GEE, it is convenient to address the difficulties encountered before writing them in the discussion section (for example, when there are neighbouring fire events, or when omission or commission errors are found in the mapping exercise). Finally, a statistical analysis of the RdNBR thresholds employed in different years/regions would be interesting, although maybe it is beyond the scope of this paper.

R: Median method for reducing image collections does not reduce noise; avoid extreme values (shadows, snow, clouds) by selecting the median value for each pixel. We applied the mosaic reducer for each image collection to get a unique image of the landscape conditions at moments

as close as possible before and after a fire event. We applied the median reducer instead when this reducer did not provide good quality pixels.

An interactive process defines the threshold as the referee mentioned. We include the sentence L 171: "This is an interactive process (fire by fire) based on visual assessment of the best RdNBR value that delimitates each individual fire scar".

We have addressed the difficulties and limitations in section 3.2 Limitations and other observations regarding the Landscape Fire Scars Database before the discussion section.

The evaluation of threshold variation in different years/regions is beyond the scope of the paper; however, the individual RdNBR is provided in the dataset for future assessment.

3)   Authors define a 10 ha limit for reconstructing the fire perimeters. This is a very big area, in excess of 100 Landsat pixels, and it would be convenient to discuss and justify this choice carefully.

R: We agreed with the reviewer comment, this is why we add an explanation that justifies the cut-off threshold of 10 ha:

L116: "The 10 ha cut-off threshold was chosen since those fires represent more than 93% of the burned area according to the CONAF official information for 1985-2018 period. In addition, small fires are usually confounded with agricultural burning, a traditional cultural practice done by Chilean farmers

4)   In the validation process, a database of 194 fire scar perimeters has been employed as a comparison source (2015-2018 years), only 78 of them reconstructed. Omitting 60% of the fire scars gives the impression that many fires are not reconstructed for various reasons (and theoretically not because of a lack of images in those years). In addition, Table 2 shows that 66.6% of fires are reconstructed (hence, omission of 33%). The above should be clearly addressed and discussed, as it is an important limitation of the database.

R: There are two main reasons to not use all the scars from Brull (2018). The first is because it was difficult to know which fire scar from Brull corresponds to the scar of our database. This is because the first one did not have the same Ids values as the base we generated (FireID) and only some of those scars shared the same other information to find it (fire name, start date, etc). To effectively compare two scars caused by the same fire, we used two different combinations of conditions using the information of the name, start date, administrative region, and season in which the fire occurred. If these conditions were not met, we preferred not to use the fire. The second reason is due to the condition that each fire in the Brull (2018) database had to be located at least 300 meters from another fire in the same season to compare only individual fire scars. These two reasons decreased the number of scars available for a fair comparison. Regarding the 33% of fires omitted, we addressed it in the first two points of section 3.2 (L301) and we also mentioned that the reason was the unavailability of satellite images due to cloud cover and the lower availability of images in developing countries and in southern hemisphere high latitude regions. This limitation of the Fire Scar Database is now addressed in point (i) Historical image

availability in the Discussion section (L351). We include a new figure 5 showing Chile's temporal trend of cloud-free pixels.

5)   It seems the accuracy of the assessment is not clearly established. When considering the 78 fires perimeters, I was expecting to see validation metrics comparing those perimeters database/ manually derived. I found only one paragraph (section 3.1) on the accuracy of the perimeters, plus the metrics shown are not clear ("global accuracy result is 0.79"). Section 2.3 refers to the methodology followed to carry out an accuracy assessment with the Closeness Index, however no results derived from this methodology for the 78 patches are shown, only some illustrative results in Figure 4. For completeness, I would also add error matrix derived metrics (user/producer accuracy or complementary omission/commission errors) in order to have similar metrics comparable to other research studies. Also convenient, I would add comparative information between your approach and coarse/resolution BA products (MCD64A1, FIRECCI products, GABAM; or the global wildfire dataset (https://doi.org/10.6084/m9.figshare.10284101) This would give an idea of the accuracy of the database to potential users.

R: The global accuracy we mentioned in the text is the mean of the calculated Closeness Index for the 78 CONAF fire scars. We make it clear in the text, showing the closeness value and adding the omission and commission errors. The formula used to calculate omission and commission error was:

Commission error = FP/(FP+TP)

Omission error = FN/(FN+TP),

FP = false positive area, FT = true positive area, and FN = false negative area.

We include the following sentence in 2.3 section (L226):

 "To assess the accuracy of our framework we include the evaluation of commission and omission error calculated as follow; commission error = FP/(FP+TP) and omission error = FN/(FN+TP), where FP is the spatial explicit false positive area of the generated fire scar compared with reference polygon of Brull (2018), FN is the false negative area and TP is the true positive area." The results of the new evaluation is at the end of 3.1 section (L309) "Finally, we found a commission error = 7% and an omission error = 28%."

Chile's Landscape Fire Scars Database is a high-resolution individual fire scar data. The global datasets have annual or monthly burned area, and as we pointed out in the introduction, Roteta et al., (2019) evaluation show a large discrepancy between local and global estimates of the burned area . Because of this and different spatial resolutions, we think a numerical comparison is beyond the scope of this research.

6)    The lack of availability of Landsat imagery is one of the sources of omission of the database. It would be interesting to follow up doing an imagery availability analysis across Chile through the years. Linked to this, the regional availability depending on the cloud cover mentioned in the text could be better contextualized.

We added the pixel and free-cloud pixel availability per year for Landsat per year in the new Figure 5.

About the data publicly available:

1)    It was straight forward to download the database - I download it without any problem.

R: We appreciate the positive review.

2)    It is reassuring the quality control described in the manuscript warrants file-concordance between fires.

R: We appreciate the positive review.

3)    It would be helpful to upload the information within this database to the GEE servers so that users may be able to use and assess it directly (for example an asset with the perimeters and severity). I would emphasise in the manuscript the reason why this database is important. For example, to me, it is not clear why post- and pre-imagery is added, and why the NBR/ RdNBR is included (I can only guess most of the people will use the perimeters/Severity associated from the process).

R: The main reason to provide the image is the transparency of the source of information. Any user can visually interpret the individual fire scar using the pre and post-image. Global datasets do not provide this information, so users can't evaluate the precision of individual fire scar data. Additionally, some users may have a specific objective as evaluating the land cover change or landscape connectivity before the fire occurrence. Providing the image can help users take advantage of the work already done.

 **[Line by line comments]**

Line 56 -> I would add an additional reference to the fire_cci BA products  (MERIS/ AVHRR / MODIS /OLCI based).

https://doi.org/10.1016/j.rse.2015.03.011 /

https://doi.org/10.1016/j.rse.2019.111493

https://doi.org/10.5194/essd-10-2015-2018

https://www.mdpi.com/2072-4292/13/21/4295

https://doi.org/10.3390/rs13214295

R: We include the references.

Line 60  (or 69)-> I would include a link to the GABAM database (Landsat, 30m) (although it is later referenced)

https://vapd.gitlab.io/post/gabam/

https://doi.org/10.3390/rs11050489

R: It is already included

Line 97 -> It would be a good addition to upload the database to GEE and share the assets.

R: The dataset is not only available for GEE users. The data is all in Pangaea repository. GEE users can easily upload after they filter what they need.

Line 106 -> Figure 1 does not correspond to the study site but to the methodological workflow

R: Reference erased

Line 112 -> the link does not point to a web page with burned area statistics but to a general web page to the CONAF.

R: Reference erased; we provide the point data. It is included in the Pangaea page as FireScar_CL_Summary_1985-2018.xlsx

Line 119-> I'm not a GEE expert but I believe GEE native language is not Javascript (it is the most popular client library because of the code editor https://code.earthengine.google.com/ )

R: To avoid confusion, we erase that part.

Line 120 -> Incorrect reference: it refers to Figure 1 and not Figure 2

R: Corrected

Line 133 ->" We use the atmospherically corrected surface reflectance and orthorectified images from Landsat 5 (1984-2013), 7 (1999-) and 8 (2013-)" The collection and GEE tag could be indicated

R: We include in the text L139: "We use the atmospherically corrected surface reflectance and orthorectified images from Landsat 5 (1984-2013) "LANDSAT/LT05/C01/T1_SR", 7 (1999-) "LANDSAT/LE07/C01/T1_SR" and 8 (2013-) "LANDSAT/LC08/C01/T1_SR"

Line 140  -> 'Pixels of snow, clouds, and cloud shadows are excluded from each image on the basis of the pixel quality band provided by Landsat.' I think these methodological details should be covered more in detail.

R: This is a standard procedure. The QA band indicates which pixels are covered by snow, clouds, or clouds shadows used as a binary mask of good and bad quality of the surface reflectance. We clarify in the text (L147).

Line 141 -> "For each image collection, we applied either the mosaic or the median reducer function to get a unique image of the landscape conditions at moments as close as posible before and after a fire event." This affirmation must be clarified. How do you get the closest burning date with the median reducer? In principle, employing the median reducer would decrease the burned signal strength

R: We include a phrase to clarify it "This can be done by sorting the image by its date, obtaining the closer good quality pixels" (L150).

Median method for reducing image collections does not reduce the burned signal strength, avoiding extreme values by selecting the median value for each pixel. We applied the mosaic reducer for each image collection to get a unique image of the landscape conditions at moments as close as possible before and after a fire event. When this reducer did not provide a good quality pixel, we applied the median reducer instead.

Line 151 -> "This index has shown better results in Mediterranean areas" Does this sentence refer to mapping burned areas or to burning severity?

R: We refer to burning severity. The index shows better results in the threshold definition between severity classes, therefore, we can expect better results between burned and unburned areas as well as we see empirically.

Line 187: "the event's severity is calculated from the RdNBR in a continuous raster format and categorized based on the ranges proposed by Miller and Thode (2007)." I think it would help writing down the ranges proposed in the manuscript

R: Now it is included in the manuscript. (L190)

Table 1: RdNBR is not fully described (square root of what in the divider?)

R: Corrected

Line 169 -> "Step (iv) involved the selection of the RdNBR index value for each wildfire that best captures the burned area based on visual interpretation. " . Please reword and clarify what this sentence mean.

R: We add the following sentence "This is an interactive process (fire by fire) based on visual assessment of the best RdNBR value that delimitates each individual fire scar." (L171)

Line 202: Please complete the Brull reference "Brull, J.: Análisis de la severidad de los incendios de magnitud de la temporada de incendios forestales 2017-2018, 2018."

R: Done

Line 206: What was the spatial distribution of the evaluation samples? If the minimum size was 200 ha and 60% were not mapped by the database (78/194), the omission of the database seems high. It would be important to calculate the omission percentage both in number and area percentage.

R: The complete answer is in Q4.

Line 210: Why not use error matrix based traditional metrics like User/producer accuracy (or the complementary Omission, commission errors)? I understand the usefulness of the polygon-based comparison due to both source and validation being polygons but I believe having and omission/commission rate would be more significant.

R: We now include the omission and commission error.

Line 256: "Using the data for all 12,250 fires recorded by CONAF between 1985 and 2018 with a burned area greater than 10 ha,". How is this information collected in CONAF? A description of the methods employed would be helpful.

R: We include an original data description in section 2.1 Data seeding

Table 2: "R is the number of reconstructed fire scars contained in our database, and UR is the number of fire scars in the database that could not be reconstructed due to the unavailability of satellite images." R and UR are not listed in the table. Are those the Yes/No columns? Please edit.

R: The referee is correct. Now is the correct reference in the text.

Line 260: A typical map of the number of Landsat scenes available across Chile would be interesting to understand changes through the years.

R: We added the pixel availability per year of Landsat in Figure 5.

Line 270: "The total number of fires >0.01 ha exhibits a positive linear relationship with the total number of fires > 10 ha also recorded by CONAF between 1985 and 2018 ($R2 = 0.86$)." I cannot establish between which two variables this relationship is performed. First, I think it would be helpful to clarify what is the CONAF dataset. Then, the slope and intercept of the regression would add valuable information about the tendency of over/ under estimation.

R: We add a more complete description of CONAF dataset in L109.

Line 272: "indicating that the distribution of the reconstructed data is regionally representative (Table 2, Figure 2)" -> please add the scatterplots as the reader may expect them.

R: We already have many images, including fig 5 by the suggestion of referee 3, so we provide the data in the table.

Line 290: Fire scar evaluation: Line 298: "Nevertheless, the global accuracy result is 0.79" This is an important result but it is not easily understood: is it the aggregated 78 'Dnorm' value? Please specify.

R: Done.

L313: Nevertheless, the global accuracy assessment derived from the Closeness Index and calculated as the mean of the individual D_norm result in a value of 0.79.

Line 300: Some of the limitations addressed here are new for the reader, I think they should be noted before in the results/methodology sections. For example, issues like having more than one fire event in neighbouring areas should have been addressed in the methodology section. The same applies for problems related to commission errors.

R: We disagree with the review. These observations come from the complete process experience and are general for the framework. This limitation section is only a "take into account" for the researcher that may be starting the process. Putting this in the methodology may be confusing because it does not apply to all cases.

Line 381: I would make clearer 5 days temporal resolution starts in 2017 with the second satellite, and that although some bands are at 10 m spatial resolution, critical bands accurate BA mapping like SWIR are at 20 m.

R: Done

Line 390 "No evident pattern associated with the latitudinal or vegetation-type change was observed in applying the threshold value to identify scars". It would be interesting to analyse the validity of the threshold values throughout various regions/years in Chile.

R: We preciate the comment, however we believe that this point is out of the scope of the paper, and should be address in a specific research.

Figure 2: Instead of using negative longitudes and latitudes, it is preferable to use South / West. For clarity, avoid adding the background shadows in the detailed maps.

R: We preciate the comment, however, the map is consistent with the data provided in the fire scar dataset.

Figure 3: I appreciate this is a plot to illustrate the computed variables, however I would include also information about the place/date of the fires illustrated.

R: We include the place/date of the fires illustrated in the caption of Fig.3.

Figure 4: I would define 'Dnorm' in the footnote for clarity

R: Done

---

## Author Response (AR2)

**Comments to the author**:**

Dear authors,

Based on my reading of your manuscript, the reviews and your responses to the reviews, your manuscript has high publication potential.

Some remaining issues (see below) need to be addressed and will be reviewed by me. Best,

Sander Veraverbeke

Major comments:

Lines 144-145: The suggested images searches pre- and -post-fire over a period of 100 days will likely result in pre- and post-fire values that represent different phenological conditions. This is a major limitation that needs to be addressed. Why not try to search for pre- and post-fire images near 'anniversary date' windows?

R: Thanks for the comment. This issue is one of the limitations when working with Landsat. In the best scenario, we can have 16 days difference between a fire event and post-fire image. However, time-lapse depends on the date and duration of the fire and the quality of post-fire image (clouds and shadows). Considering this, we select the best pixels at moments as close as possible after a fire event sorting the image collection by the date with a maximum limit of 100 days. We consider the anniversary date a possibility, but we discarded it because of vegetation recovery or tree or crop replanting in productive lands. In addition, in Chile there is still salvage logging a legal management action after fire, that could severely affect the post-fire land cover spectral response. Considering that the maximum time between the event and the next image could be a limitation for some places, we make explicit in the text that this decision could be taken locally.

L 147: However, the definition of the maximum period of time must be chosen, taking into account local phenology, vegetation recovery, or landscape dynamics, aspects that could change the spectral response of the land surface after a fire.

Lines 171-172: the interactive visual interpretation process to decide fire-by-fire on a RdNBR threshold is a serious limitation. These reviewer comments have not been sufficiently addressed and I highly recommend thinking about automating the process. Could (R)dNBR histograms of known unburned pixels in your time series be helpful to decide on objective thresholds? Possibly, such a threshold could be spatially and temporally variable.

R: Thanks for the comment. We are now working on automatizing this process; however, we need an initial visual interpretation of this threshold because it changes depending on the vegetation type affected by fire, date, severity, and other conditions until unknown for Chile. The visually interpreted boundary of the fire gives us the "ground truth" for validating our future development. Comparing the distribution of RdNBR in a burned and unburned site is a good start, but we still need this validation data done by the interpretation from the human eyes. Deep learning technics as an Artificial intelligent model is a promising approach that tries to imitate our brain's image interpretation. However, the models still need training images obtained from this "ground truth" data.

Lines 210-211. The reduction from 194 to 78 fire scars needs to be justified (as requested by one of the reviewers). Please explicitly state the reasons of the omissions, and the number of omitted fires for each reason.

R: We clarify in the text.

L212: The author generated 194 fire scars, of which 78 coincided with two criteria for making comparisons: (i) individual fire scar must be at a distance of at least 300 m of another reported fire by Brull (2018), and (ii) the fire have the same name, start date and control of our seed data to avoid confusion. From 194 fire scars collected by Brull (2018), 107 were within 300 m from another reported fire, and in nine cases, the name and dates did not match.

Minor comments:

Lines 64-70. Would be good to include some recent efforts here based on Sentinel-2 data, for example:

Ramo, Ruben, et al. "African burned area and fire carbon emissions are strongly impacted by small fires undetected by coarse resolution satellite data." Proceedings of the National Academy of Sciences 118.9 (2021): e2011160118.

Glushkov, Igor, et al. "Spring fires in Russia: Results from participatory burned area mapping with Sentinel-2 imagery." Environmental Research Letters 16.12 (2021): 125005.

R: We included.

Line 104: Km2 should be km2

**R: Corrected**

Line 119: please remove 's for database

**R: Corrected**

Lines 403-404: 'Different burned-land covers may have variable RdNBR values, but this relationship does not figure among the objectives for the present database development.' Somewhat awkward statement, I suggest removing.

**R: Removed**

Line 430: 'FIRMS': you are referring to the VIIRS active fire data here, please rephrase.

**R: Corrected**

Figure 4. Lat/lon labels are too small to read. I recommended adding them with larger font size to the top left map figure, and removing them from the other maps. I also concur with reviewer 3 that it is recommended to use South & West in the latitude & longitude labeling, rather than using negative values.

R: Corrected. We change lat long for South and West for Fig 2 and 5. Fig. 4 doesn't have coordinates.